# A 3D Generative Model for Structure-Based Drug Design

**Shitong Luo**
HeliXon Research
luost@helixon.com
luost26@gmail.com

**Jiaqi Guan**
University of Illinois Urbana-Champaign
jiaqi@illinois.edu

**Jianzhu Ma**
Peking University
majianzhu@pku.edu.cn

**Jian Peng**
University of Illinois Urbana-Champaign
jianpeng@illinois.edu

## Abstract

We study a fundamental problem in structure-based drug design — generating molecules that bind to specific protein binding sites. While we have witnessed the great success of deep generative models in drug design, the existing methods are mostly string-based or graph-based. They are limited by the lack of spatial information and thus unable to be applied to structure-based design tasks. Particularly, such models have no or little knowledge of how molecules interact with their target proteins exactly in 3D space. In this paper, we propose a 3D generative model that generates molecules given a designated 3D protein binding site. Specifically, given a binding site as the 3D context, our model estimates the probability density of atom's occurrences in 3D space — positions that are more likely to have atoms will be assigned higher probability. To generate 3D molecules, we propose an auto-regressive sampling scheme — atoms are sampled sequentially from the learned distribution until there is no room for new atoms. Combined with this sampling scheme, our model can generate valid and diverse molecules, which could be applicable to various structure-based molecular design tasks such as molecule sampling and linker design. Experimental results demonstrate that molecules sampled from our model exhibit high binding affinity to specific targets and good drug properties such as drug-likeness even if the model is not explicitly optimized for them.

## 1 Introduction

Designing molecules that bind to a specific protein binding site, also known as structure-based drug design, is one of the most challenging tasks in drug discovery [2]. Searching for suitable molecule candidates *in silico* usually involves massive computational efforts because of the enormous space of synthetically feasible chemicals [22] and conformational degree of freedom of both compound and protein structures [11].

In recent years, we have witnessed the success of machine learning approaches to problems in drug design, especially on molecule generation. Most of these approaches use deep generative models to propose drug candidates by learning the underlying distribution of desirable molecules. However, most of such methods are generally SMILES/string-based [10, 17] or graph-based [18, 19, 13, 14]. They are limited by the lack of spatial information and unable to perceive how molecules interact with proteins in 3D space. Hence, these methods are not applicable to generating molecules that fit to a specific protein structure which is also known as the drug target. Another line of work studies

generating molecules directly in 3D space [8, 28, 29, 20, 30, 15]. Most of them [8, 28, 29] can only handle very small organic molecules, not sufficient to generate drug-scale molecules which usually contain dozens of heavy atoms. [20] proposes to generate voxelized molecular images and use a post-processing algorithm to reconstruct molecular structures. Though this method could produce drug-scale molecules for specific protein pockets, the quality of the sampling is heavily limited by voxelization. Therefore, generating high-quality drug molecules for specific 3D protein binding sites remains challenging.

In this work, we propose a 3D generative model to approach this task. Specifically, we aim at modeling the distribution of atom occurrence in the 3D space of the binding site. Formally, given a binding site $\mathcal{C}$ as input, we model the distribution $p(e, \boldsymbol{r}|\mathcal{C})$, where $\boldsymbol{r} \in \mathbb{R}^3$ is an arbitrary 3D coordinate and $e$ is atom type. To realize this distribution, we design a neural network architecture which takes as input a query 3D coordinate $\boldsymbol{r}$, conditional on the 3D context $\mathcal{C}$, and outputs the probability of $\boldsymbol{r}$ being occupied by an atom of a particular chemical element. In order to ensure the distribution is equivariant to $\mathcal{C}$'s rotation and translation, we utilize rotationally invariant graph neural networks to perceive the context of each query coordinate.

Despite having a neural network to model the distribution of atom occurrence $p(e, \boldsymbol{r}|\mathcal{C})$, how to generate *valid* and *diverse* molecules still remains technically challenging, mainly for the following two reasons: First, simply drawing *i.i.d.* samples from the distribution $p(e, \boldsymbol{r}|\mathcal{C})$ does not yield valid molecules because atoms within a molecule are not independent of each other. Second, a desirable sampling algorithm should capture the multi-modality of the feasible chemical space, *i.e.* it should be able to generate a diverse set of desired molecules given a specific binding context. To tackle the challenge, we propose an auto-regressive sampling algorithm. In specific, we start with a context consisting of only protein atoms. Then, we iteratively sample one atom from the distribution at each step and add it to the context to be used in the next step, until there is no room for new atoms. Compared to other recent methods [20, 23], our auto-regressive algorithm is simpler and more advantageous. It does not rely on post-processing algorithms to infer atom placements from density. More importantly, it is capable of multi-modal sampling by the nature of auto-regressive, avoiding additional latent variables via VAEs [16] or GANs [9] which would bring about extra architectural complexity and training difficulty.

We conduct extensive experiments to evaluate our approach. Quantitative and qualitative results show that: (1) our method is able to generate diverse drug-like molecules that have high binding affinity to specific targets based on 3D structures of protein binding sites; (2) our method is able to generate molecules with fairly high drug-likeness score (QED) [4] and synthetic accessibility score (SA) [6] even if the model is not specifically optimized for them; (3) in addition to molecule generation, the proposed method is also applicable to other relevant tasks such as linker design.

## 2    Related Work

**SMILES-Based and Graph-Based Molecule Generation**    Deep generative models have been prevalent in molecule design. The overall idea is to use deep generative models to propose molecule candidates by learning the underlying distribution of desirable molecules. Existing works can be roughly divided into two classes — string-based and graph-based. String-based methods represent molecules as linear strings, e.g. SMILES strings [34], making a wide range of language modeling tools readily applicable. For example, [5, 10, 26] utilize recurrent neural networks to learn a language model of SMILES strings. However, string-based representations fail to capture molecular similarities, making it a sub-optimal representation for molecules [13]. In contrast, graph representations are more natural, and graph-based approaches have drawn great attention. The majority of graph-based models generate molecules in an auto-regressive fashion, i.e., adding atoms or fragments sequentially, which could be implemented based upon VAEs [13], normalizing flows [27], reinforcement learning [35, 14], etc. Despite the progress made in string-based and graph-based approaches, they are limited by the lack of spatial information and thus unable to be directly applied to structure-based drug design tasks [2]. Specifically, as 1D/2D-based methods, they are unable to perceive how molecules interact with their target proteins exactly in 3D space.

**Molecule Generation in 3D Space**    There has been another line of methods that generate molecules directly in 3D space. [8] proposes an auto-regressive model which takes a partially generated molecule as input and outputs the next atom's chemical element and the distances to previous atoms and places

the atoms in the 3D space according to the distance constraints. [28, 29] approach this task via reinforcement learning by generating 3D molecules in a sequential way. Different from the previous method[8], they mainly rely on a reward function derived from the potential energy function of atomic systems. These works could generate *realistic* 3D molecules. However, they can only handle small organic molecules, not sufficient to generate drug-scale molecules which usually contain dozens of heavy atoms.

[20, 23] propose a non-autoregressive approach to 3D molecular generation which is able to generate *drug-scale* molecules. It represents molecules as 3D images by voxelizing molecules onto 3D meshgrids. In this way, the molecular generation problem is transformed into an image generation problem, making it possible to leverage sophisticated image generation techniques. In specific, it employs convolutional neural network-based VAEs [16] or GANs [9] to generate such molecular images. It also attempts to fuse the binding site structures into the generative network, enabling the model to generate molecules for designated binding targets. In order to reconstruct the molecular structures from images, it leverages a post-processing algorithm to search for atom placements that best fit the image. In comparison to previous methods which can only generate small 3D molecules, this method can generate drug-scale 3D molecules. However, the quality of its generated molecules is not satisfying because of the following major limitations. First, it is hardly scalable to large binding pockets, as the number of voxels grows cubically to the size of the binding site. Second, the resolution of the 3D molecular images is another bottleneck that significantly limits the precision due to the same scalability issue. Last, conventional CNNs are not rotation-equivariant, which is crucial for modeling molecular systems [25].

## 3 Method

Our goal is to generate a set of atoms that is able to form a valid drug-like molecule fitting to a specific binding site. To this end, we first present a 3D generative model in Section 3.1 that predicts the probability of atom occurrence in 3D space of the binding site. Second, we present in Section 3.2 the auto-regressive sampling algorithm for generating valid and multi-modal molecules from the model. Finally, in Section 3.3, we derive the training objective, by which the model learns to predict where should be placed and atoms and what type of atom should be placed.

### 3.1 3D Generative Model Design

A binding site can be defined as a set of atoms $\mathcal{C} = \{(\boldsymbol{a}_i, \boldsymbol{r}_i)\}_{i=1}^{N_b}$, where $N_b$ is the number of atoms in the binding site, $\boldsymbol{a}_i$ is the $i$-th atom's attributes such as chemical element, belonging amino acid, etc., and $\boldsymbol{r}_i$ is its 3D coordinate. To generate atoms in the binding site, we consider modeling the probability of atom occurring at some position $\boldsymbol{r}$ in the site. Formally, this is to model the density $p(e|\boldsymbol{r}, \mathcal{C})$, where $\boldsymbol{r} \in \mathbb{R}^3$ is an arbitrary 3D coordinate, and $e \in \mathcal{E} = \{\text{H}, \text{C}, \text{O}, \ldots\}$ is the chemical element. Intuitively, this density can be interpreted as a classifier that takes as input a 3D coordinate $\boldsymbol{r}$ conditional on $\mathcal{C}$ and predicts the probability of $\boldsymbol{r}$ being occupied by an atom of type $e$.

To model $p(e|\boldsymbol{r}, \mathcal{C})$, we devise a model consisting of two parts: **Context Encoder** learns the representation of each atom in the context $\mathcal{C}$ via graph neural networks. **Spatial Classifier** takes as input a query position $\boldsymbol{r}$, then aggregates the representation of contextual atoms nearby it, and finally predicts $p(e|\boldsymbol{r}, \mathcal{C})$. The implementation of these two parts is detailed as follows.

**Context Encoder**    The purpose of the context encoder is to extract information-rich representations for each atom in $\mathcal{C}$. We assume a desirable representation should satisfy two properties: (1) **context-awareness**: the representation of an atom should not only encode the property of the atom itself, but also encode its context. (2) **rotational and translational invariance**: since the physical and biological properties of the system do not change according to rigid transforms, the representations that reflect these properties should be invariant to rigid transforms as well. To this end, we employ rotationally and translationally invariant graph neural networks [25] as the backbone of the context encoder, described as follows.

First of all, since there is generally no natural topology in $\mathcal{C}$, we construct a $k$-nearest-neighbor graph based on inter-atomic distances, denoted as $\mathcal{G} = \langle \mathcal{C}, \boldsymbol{A} \rangle$, where $\boldsymbol{A}$ is the adjacency matrix. We also denote the $k$-NN neighborhood of atom $i$ as $N_k(\boldsymbol{r}_i)$ for convenience. The context encoder will take $\mathcal{G}$ as input and output structure-aware node embeddings.

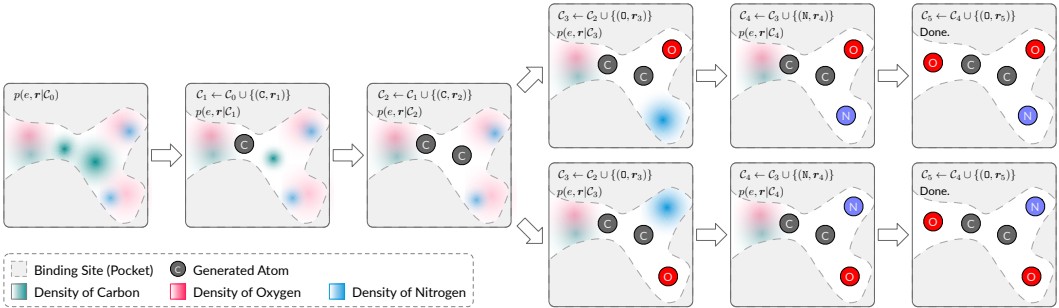

Figure 1: An illustration of the sampling process. Atoms are sampled sequentially. The probability density changes as we place new atoms. The sampling process naturally diverges, leading to different samples.

The first layer of the encoder is a linear layer. It maps atomic attributes $\{a_i\}$ to initial embeddings $\{h_i^{(0)}\}$. Then, these embeddings along with the graph structure $A$ are fed into $L$ message passing layers. Specifically, the formula of message passing takes the form:

$$h_i^{(\ell+1)} = \sigma \left( W_0^\ell h_i^{(\ell)} + \sum_{j \in N_k(r_i)} W_1^\ell w(d_{ij}) \odot W_2^\ell h_j^{(\ell)} \right), \tag{1}$$

where $w(\cdot)$ is a weight network and $d_{ij}$ denotes the distance between atom $i$ and atom $j$. The formula is similar to continuous filter convolution [25]. Note that, the weight of message from $j$ to $i$ depends only on $d_{ij}$, ensuring its invariance to rotation and translation. Finally, we obtain $\{h_i^{(L)}\}$ a set of embeddings for each atom in $\mathcal{C}$.

**Spatial Classifier**   The spatial classifier takes as input a query position $r \in \mathbb{R}^3$ and predicts the type of atom occupying $r$. In order to make successful predictions, the model should be able to perceive the context around $r$. Therefore, the first step of this part is to aggregate atom embeddings from the context encoder:

$$v = \sum_{j \in N_k(r)} W_0 w_{\text{aggr}}(\|r - r_j\|) \odot W_1 h_j^{(L)}, \tag{2}$$

where $N_k(r)$ is the $k$-nearest neighborhood of $r$. Note that we weight different embedding using the weight network $w_{\text{aggr}}(\cdot)$ according to distances because it is necessary to distinguish the contribution of different atoms in the context. Finally, in order to predict $p(e|r, \mathcal{C})$, the aggregated feature $v$ is then passed to a classical multi-layer perceptron classifier:

$$c = \text{MLP}(v), \tag{3}$$

where $c$ is the non-normalized probability of chemical elements. The estimated probability of position $r$ being occupied by atom of type $e$ is:

$$p(e|r, \mathcal{C}) = \frac{\exp\left(c[e]\right)}{1 + \sum_{e' \in \mathcal{E}} \exp\left(c[e']\right)}, \tag{4}$$

where $\mathcal{E}$ is the set of possible chemical elements. Unlike typical classifiers that apply softmax to $c$, we make use of the extra degree of freedom by adding 1 to the denominator, so that the probability of "nothing" can be expressed as:

$$p(\texttt{Nothing}|r, \mathcal{C}) = \frac{1}{1 + \sum \exp(c[e']))}. \tag{5}$$

## 3.2   Sampling

Sampling a molecule amounts to generating a set of atoms $\{(e_i, r_i)\}_{i=1}^{N_a}$. However, formulating an effective sampling algorithm is non-trivial because of the following three challenges. First, we have to define the joint distribution of $e$ and $r$, i.e. $p(e, r|\mathcal{C})$, from which we can jointly sample an atom's

chemical element and its position. Second, notice that simply drawing i.i.d. samples from $p(e, \boldsymbol{r}|\mathcal{C})$ doesn't make sense because atoms are clearly not independent of each other. Thus, the sampling algorithm should be able to attend to the dependencies between atoms. Third, the sampling algorithm should produce multi-modal samples. This is important because in reality there is usually more than one molecule that can bind to a specific target.

In the following, we first define the joint distribution $p(e, \boldsymbol{r}|\mathcal{C})$. Then, we present an auto-regressive sampling algorithm to tackle the second and the third challenges.

**Joint Distribution**  We define the joint distribution of coordinate $\boldsymbol{r}$ and atom type $e$ using Eq.4:

$$p(e, \boldsymbol{r}|\mathcal{C}) = \frac{\exp\left(\boldsymbol{c}[e]\right)}{Z}, \tag{6}$$

where $Z$ is an unknown normalizing constant and $\boldsymbol{c}$ is a function of $\boldsymbol{r}$ and $\mathcal{C}$ as defined in Eq.3. Though $p(e, \boldsymbol{r})$ is a non-normalized distribution, drawing samples from it would be efficient because the dimension of $\boldsymbol{r}$ is only 3. Viable sampling methods include Markov chain Monte Carlo (MCMC) or discretization.

**Auto-Regressive Sampling**  We sample a molecule by progressively sampling one atom at each step. In specific, at step $t$, the context $\mathcal{C}_t$ contains not only protein atoms but also $t$ atoms sampled beforehand. Sampled atoms in $\mathcal{C}_t$ are treated equally as protein atoms in the model, but they have different attributes in order to differentiate themselves from protein atoms. Then, the $(t+1)$-th atom will be sampled from $p(e, \boldsymbol{r}|\mathcal{C}_t)$ and will be added to $\mathcal{C}_t$, leading to the context for next step $\mathcal{C}_{t+1}$. The sampling process is illustrated in Figure 1. Formally, we have:

$$\begin{aligned} (e_{t+1}, \boldsymbol{r}_{t+1}) &\sim p(e, \boldsymbol{r}|\mathcal{C}_t), \\ \mathcal{C}_{t+1} &\leftarrow \mathcal{C}_t \cup \{(e_{t+1}, \boldsymbol{r}_{t+1})\}. \end{aligned} \tag{7}$$

To determine when the auto-regressive sampling should stop, we employ an auxiliary network. The network takes as input the embedding of previously sampled atoms, and classifies them into two categories: frontier and non-frontier. If all the existing atoms are non-frontier, which means there is no room for more atoms, the sampling will be terminated. Finally, we use OpenBabel [21, 20] to obtain bonds of generated structures.

In summary, the proposed auto-regressive algorithm succeeds to settle the aforementioned two challenges. First, the model is aware of other atoms when placing new atoms, thus being able to consider the dependencies between them. Second, auto-regressive sampling is a stochastic process. Its sampling path naturally diverges, leading to diverse samples.

### 3.3  Training

As we adopt auto-regressive sampling strategies, we propose a cloze-filling training scheme — at training time, a random portion of the target molecule is masked, and the network learns to predict the masked part from the observable part and the binding site. This emulates the sampling process where the model can only observe partial molecules. The training loss consists of three terms described below.

First, to make sure the model is able to predict positions that actually have atoms (positive positions), we include a binary cross entropy loss to contrast positive positions against negative positions:

$$L_{\text{BCE}} = -\mathbb{E}_{\boldsymbol{r} \sim p_+}\left[\log\left(1 - p(\texttt{Nothing}|\boldsymbol{r}, \mathcal{C})\right)\right] - \mathbb{E}_{\boldsymbol{r} \sim p_-}\left[\log p(\texttt{Nothing}|\boldsymbol{r}, \mathcal{C})\right]. \tag{8}$$

Here, $p_+$ is a positive sampler that yields coordinates of masked atoms. $p_-$ is a negative sampler that yields random coordinates in the ambient space. $p_-$ is empirically defined as a Gaussian mixture model containing $|\mathcal{C}|$ components centered at each atom in $\mathcal{C}$. The standard deviation of each component is set to 2Å in order cover to the ambient space. Intuitively, the first term in Eq.8 increases the likelihood of atom placement for positions that should get an atom. The second term decreases the likelihood for other positions.

Second, our model should be able to predict the chemical element of atoms. Hence, we further include a standard categorical cross entropy loss:

$$L_{\text{CAT}} = -\mathbb{E}_{(e, \boldsymbol{r}) \sim p_+}\left[\log p(e|\boldsymbol{r}, \mathcal{C})\right]. \tag{9}$$

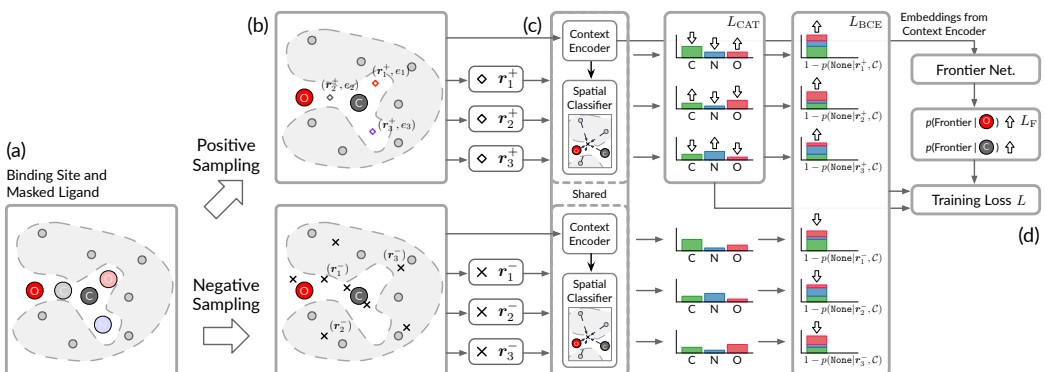

Figure 2: (a) A portion of the molecule is masked. (b) Positive coordinates are drawn from the masked atoms' positions and negative coordinates are drawn from the ambient space. (c) Both positive and negative coordinates are fed into the model. The model predicts the probability of atom occurrence at the coordinates. (d) Training losses are computed based on the discrepancy between predicted probabilities and ground truth.

Third, as introduced in Section 3.2, the sampling algorithm requires a frontier network to tell whether the sampling should be terminated. This leads to the last term — a standard binary cross entropy loss for training the frontier network:

$$L_{\mathrm{F}} = \sum_{i \in \mathcal{F} \subseteq \mathcal{C}} \log \sigma(F(\boldsymbol{h}_i)) + \sum_{i \notin \mathcal{F} \subseteq \mathcal{C}} \log(1 - \sigma(F(\boldsymbol{h}_i))), \tag{10}$$

where $\mathcal{F}$ is the set of frontier atoms in $\mathcal{C}$, $\sigma$ is the sigmoid function, and $F(\cdot)$ is the frontier network that takes atom embedding as input and predicts the logit probability of the atom being a frontier. During training, an atom is regarded as a frontier if and only if (1) the atom is a part of the target molecule, and (2) at least one of its bonded atom is masked.

Finally, by summing up $L_{\mathrm{BCE}}$, $L_{\mathrm{CAT}}$, and $L_{\mathrm{F}}$, we obtain the full training loss $L = L_{\mathrm{BCE}} + L_{\mathrm{CAT}} + L_{\mathrm{F}}$. The full training process is illustrated in Figure 2.

## 4 Experiments

We evaluate the proposed method on two relevant structure-based drug design tasks: (1) **Molecule Design** is to generate molecules for given binding sites (Section 4.1), and (2) **Linker Prediction** is to generate substructures to link two given fragments in the binding site. (Section 4.2). Below, we describe common setups shared across tasks. Detailed task-specific setups are provided in each subsection.

**Data**    We use the CrossDocked dataset [7] following [20]. The dataset originally contains 22.5 million docked protein-ligand pairs at different levels of quality. We filter out data points whose binding pose RMSD is greater than 1Å, leading to a refined subset consisting of 184,057 data points. We use mmseqs2 [31] to cluster data at 30% sequence identity, and randomly draw 100,000 protein-ligand pairs for training and 100 proteins from remaining clusters for testing.

**Model**    We trained a universal model for all the tasks. The number of message passing layers in context encoder $L$ is 6, and the hidden dimension is 256. We train the model using the Adam optimizer at learning rate 0.0001. Other details about model architectures and training parameters are provided in the supplementary material and the open source repository: `https://github.com/luost26/3D-Generative-SBDD`.

| Metric | | liGAN | Ours | Ref |
|--------|------|-------|------|-----|
| Vina Score (kcal/mol, ↓) | Avg. | -6.144 | **-6.344** | -7.158 |
| | Med. | -6.100 | **-6.200** | -6.950 |
| QED (↑) | Avg. | 0.371 | **0.525** | 0.484 |
| | Med. | 0.369 | **0.519** | 0.469 |
| SA (↑) | Avg. | 0.591 | **0.657** | 0.733 |
| | Med. | 0.570 | **0.650** | 0.745 |
| High Affinity (%, ↑) | Avg. | 23.77 | **29.09** | - |
| | Med. | 11.00 | **18.50** | - |
| Diversity (↑) | Avg. | 0.655 | **0.720** | - |
| | Med. | 0.676 | **0.736** | - |

Table 1: Mean and median values of the four metrics on generation quality. (↑) indicates higher is better. (↓) indicates lower is better.

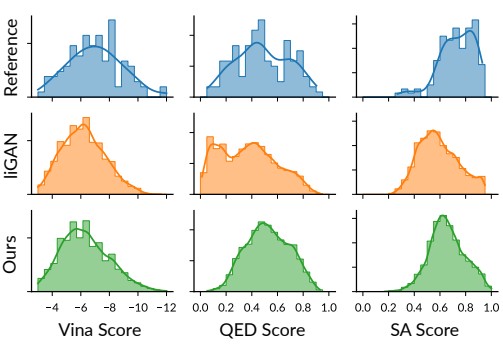

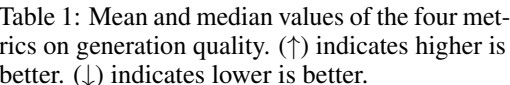

Figure 3: Distributions of Vina, QED, and SA scores over all the generated molecules.

## 4.1 Molecule Design

In this task, we generate molecules for specific binding sites with our model and baselines. The input to models are binding sites extracted from the proteins in the testing set. We sample 100 unique molecules for each target.

**Baselines**    We compare our approach with the state-of-the-art baseline liGAN [20]. liGAN is based on conventional 3D convolutional neural networks. It generates voxelized molecular images and relies on a post-processing algorithm to reconstruct the molecule from the generated image.

**Metrics**    We evaluate the quality of generated molecules from three main aspects: (1) **Binding Affinity** measures how well the generated molecules fit the binding site. We use Vina [33, 1] to compute the binding affinity (*Vina Score*). Before feeding the molecules to Vina, we employ the universal force fields (UFF) [24] to refine the generated structures following [20]. (2) **Drug Likeness** reflects how much a molecule is like a drug. We use *QED* score [4] as the metric for drug-likeness. (3) **Synthesizability** assesses the ease of synthesis of generated molecules. We use normalized *SA* score [6, 35] to measure molecules' synthesizability.

In order to evaluate the generation quality and diversity for each binding site, we define two additional metrics: (1) **Percentage of Samples with High Affinity**, which measures the percentage of a binding site's generated molecules whose binding affinity is *higher than or equal to* the reference ligand. (2) **Diversity** [14], which measures the diversity of generated molecules for a binding site. It is calculated by averaging pairwise Tanimoto similarities [3, 32] over Morgan fingerprints among the generated molecules of a target.

**Results**    We first calculate Vina Score, QED, and SA for each of the generated molecules. Figure 3 presents the histogram of these three metrics and Table 1 shows the mean and median values of them over all generated molecules. For each binding site, we further calculate Percentage of Samples with High Affinity and Diversity. We report their mean and median values in the bottom half of Table 1. From the quantitative results, we find that in general, our model is able to discover diverse molecules that have higher binding affinity to specific targets. Besides, the generated molecules from our model also exhibit other desirable properties including fairly high drug-likeness and synthesizability. When compared to the CNN baseline liGAN [20], our method achieves clearly better performance on all metrics, especially on the drug-likeness score QED, which indicates that our model produces more realistic drug-like molecules.

To better understand the results, we select two binding sites in the testing set and visualize their top affinity samples for closer inspection. The top row of Figure 4 is the first example (PDB ID:2hcj). The average QED and SA scores of the generated molecules for this target are 0.483 and 0.663

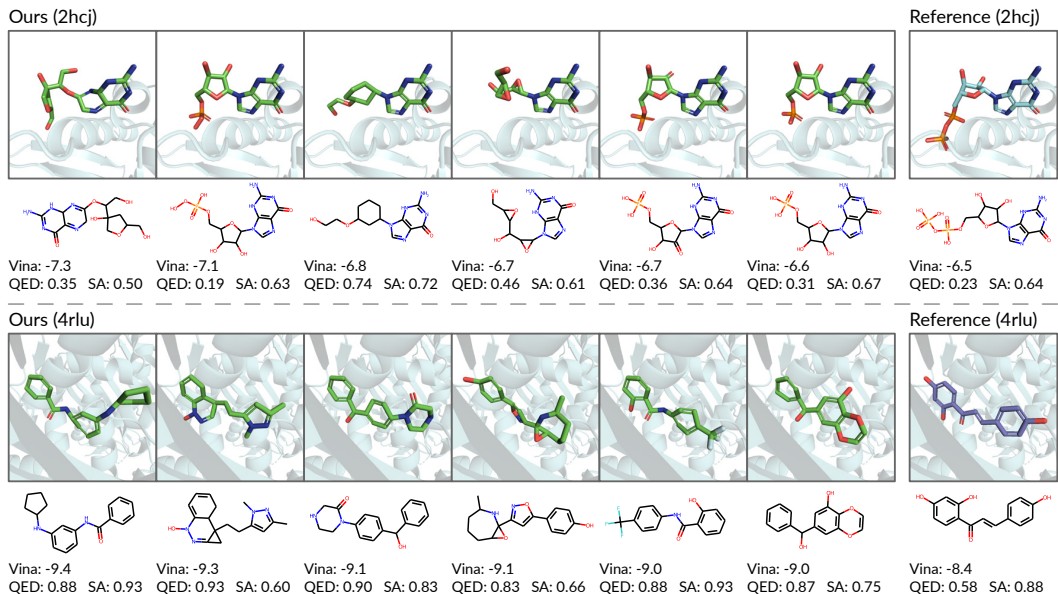

Figure 4: Generated molecules with top binding affinity and the reference molecule for two representative binding sites. Lower Vina score indicates higher binding affinity.

respectively, around the median of these two scores. 8% of the generated molecules have higher binding affinity than the reference molecule, below the median 18.5%. The second example (PDB ID:4rlu) is shown in the bottom row. The average QED and SA scores are 0.728 and 0.785, and 18% of sampled molecules achieve higher binding affinity. From these two examples in Figure 4, we can see that the generated molecules have overall structures similar to the reference molecule and they share some common important substructures, which indicates that the generated molecules fit into the binding site as well as the reference one. Besides, the top affinity molecules generally achieve QED and SA score comparable to or even higher than the reference molecule, which reflects that the top affinity molecules not only fit well into the binding site but also exhibit desirable quality. In conclusion, the above two representative cases evidence the model's ability to generate drug-like and high binding affinity molecules for designated targets.

## 4.2 Linker Prediction

Linker prediction is to build a molecule that incorporates two given disconnected fragments in the context of a binding site [12]. Our model is capable of linker design without any task-specific adaptation or re-training. In specific, given a binding site and some fragments as input, we compose the initial context $\mathcal{C}_0$ containing both the binding site and the fragments. Then, we run the auto-regressive sampling algorithm to sequentially add atoms until the molecule is complete.

Table 2: Performance of linker prediction.

| Metric | | DeLinker | **Ours** |
|---|---|---|---|
| Similarity (↑) | Avg. | 0.612 | **0.701** |
| | Med. | 0.600 | **0.722** |
| Recovered (%, ↑) | | 40.00 | **48.33** |
| Vina Score | Avg. | -8.512 | -8.603 |
| (kcal/mol, ↓) | Med. | -8.576 | -8.575 |

**Data Preparation** Following [12], we construct fragments of molecules in the testing set by enumerating possible double-cuts of acyclic single bonds. The pre-processing results in 120 data points in total. Each of them consists of two disconnected molecule fragments.

**Baselines** We compare our model with DeLinker [12]. Despite that DeLinker incorporates some 3D information, it is still a graph-based generative model. In contrast, our method operates fully in 3D space and thus is able to fully utilize the 3D context.

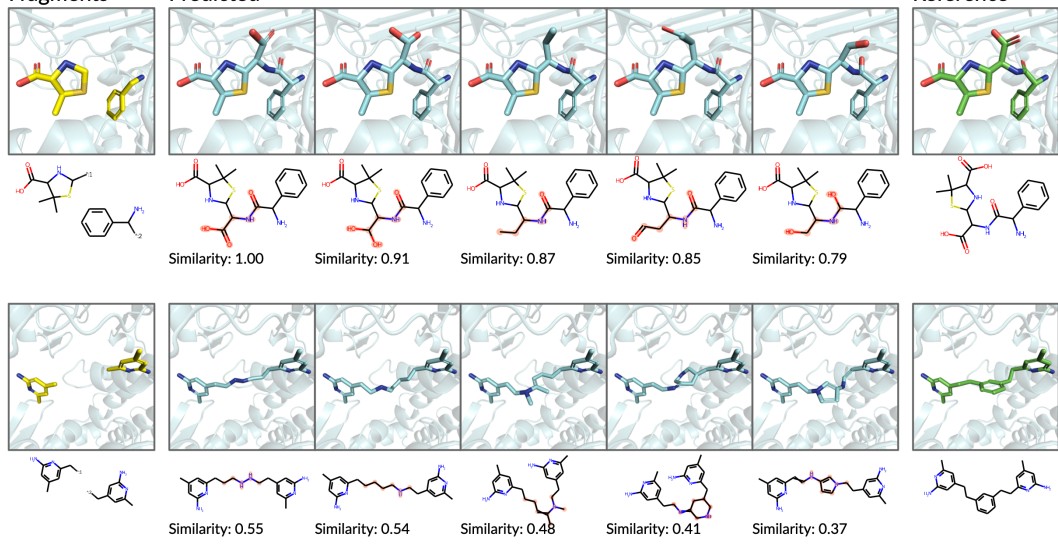

Figure 5: Two example of linker prediction. Atoms highlighted in red are predicted linkers.

**Metrics**    We assess the generated molecules from fragments with four main metrics: (1) **Similarity**: We use Tanimoto Similarity [32, 3] over Morgan fingerprints [14] to measure the similarity between the molecular graphs of generated molecule and the reference molecule. (2) **Percentage of Recovered Molecules**: We say a test molecule is recovered if the model is able to generate a molecule that perfectly matches it (Similarity = 1.0). We calculate the percentage of test molecules that are recovered by the model. (3) **Binding Affinity**: We use Vina [1, 33] to compute the the generated molecules' binding affinity to the target.

**Results**    For each data point, we use our model and DeLinker to generate 100 molecules. We first calculate the average similarity for each data point and report their overall mean and median values. Then, we calculate the percentage of test molecules that are successfully recovered by the model. Finally, we use Vina to evaluate the generated molecules' binding affinity. These results are summarized in Table 2. As shown in the table, when measured by Vina score, our proposed method's performance is on par with the graph-based baseline DeLinker. However, our method clearly outperforms DeLinker on Similarity and Percentage of Recovery, suggesting that our method is able to link fragments in a more realistic way. In addition, we present two examples along with 5 generated molecules at different similarities in Figure 5. The example demonstrates the model's ability to generate suitable linkers.

## 5    Conclusions and Discussions

In this paper, we propose a new approach to structure-based drug design. In specific, we design a 3D generative model that estimates the probability density of atom's occurrences in 3D space and formulate an auto-regressive sampling algorithm. Combined with the sampling algorithm, the model is able to generate drug-like molecules for specific binding sites. By conducting extensive experiments, we demonstrate our model's effectiveness in designing molecules for specific targets. Though our proposed method achieves reasonable performance in structure-based molecule design, there is no guarantee that the model always generates valid molecules successfully. To build a more robust and useful model, we can consider incorporating graph representations to building 3D molecules as future work, such that we can leverage on sophisticated techniques for generating valid molecular graphs such as valency check [35] and property optimization [14].

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
