# Supplementary Material

## A    Additional Results

### A.1    Molecule Design

We present more examples of generated molecules by our method and the CNN baseline liGAN. We select 6 molecules with highest binding affinity for each method and each binding site. The 3 additional binding sites are selected randomly from the testing set. By comparing the samples from two methods, we can find that the 3D molecules generated by our method are generally more realistic, while molecules generated by the baseline have more erroneous structures, such as bonds that are too short and angles that are too sharp. Besides, molecules generated by our method are more diverse, while the 3D atom configurations generated by the baseline are often similar. More importantly, our model can generate novel molecules that are obviously different from the reference molecule and achieve higher binding affinity. To summarize, these examples evidence the proposed model's good performance in terms of drug-likeness, diversity, and binding affinity.

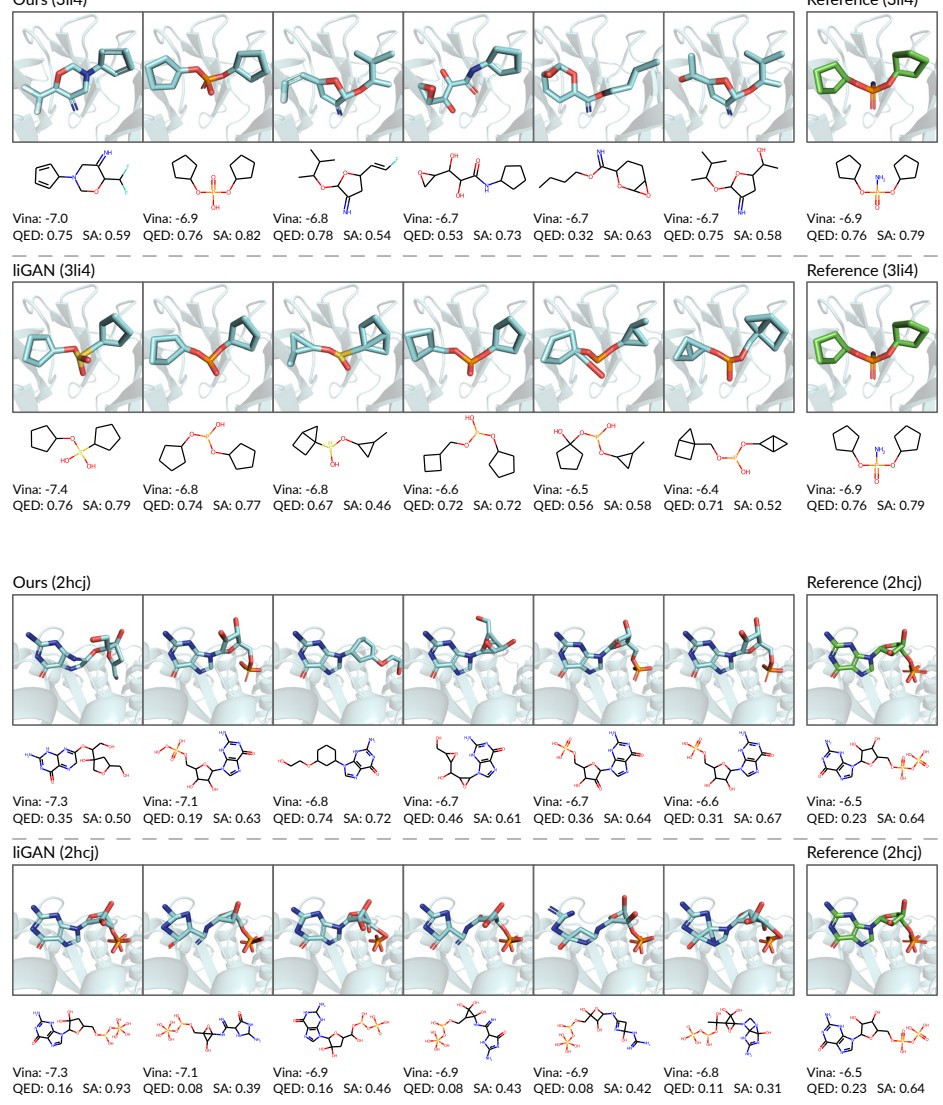

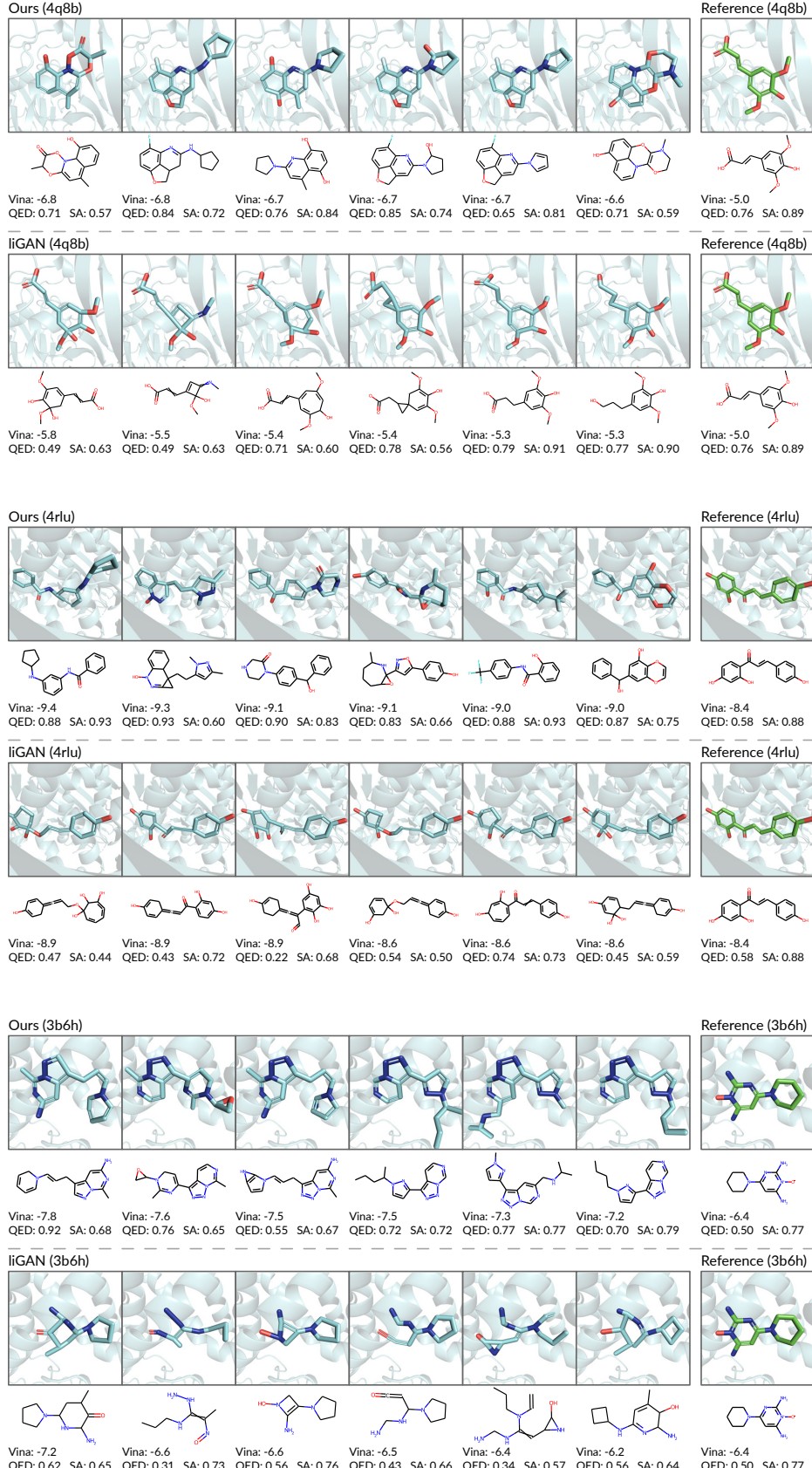

Ours (4q8b) — Vina: -6.8 QED: 0.71 SA: 0.57 / Vina: -6.8 QED: 0.84 SA: 0.72 / Vina: -6.7 QED: 0.76 SA: 0.84 / Vina: -6.7 QED: 0.85 SA: 0.74 / Vina: -6.7 QED: 0.65 SA: 0.81 / Vina: -6.6 QED: 0.71 SA: 0.59 — Reference (4q8b) Vina: -5.0 QED: 0.76 SA: 0.89

liGAN (4q8b) — Vina: -5.8 QED: 0.49 SA: 0.63 / Vina: -5.5 QED: 0.49 SA: 0.63 / Vina: -5.4 QED: 0.71 SA: 0.60 / Vina: -5.4 QED: 0.78 SA: 0.56 / Vina: -5.3 QED: 0.79 SA: 0.91 / Vina: -5.3 QED: 0.77 SA: 0.90 — Reference (4q8b) Vina: -5.0 QED: 0.76 SA: 0.89

Ours (4rlu) — Vina: -9.4 QED: 0.88 SA: 0.93 / Vina: -9.3 QED: 0.93 SA: 0.60 / Vina: -9.1 QED: 0.90 SA: 0.83 / Vina: -9.1 QED: 0.83 SA: 0.66 / Vina: -9.0 QED: 0.88 SA: 0.93 / Vina: -9.0 QED: 0.87 SA: 0.75 — Reference (4rlu) Vina: -8.4 QED: 0.58 SA: 0.88

liGAN (4rlu) — Vina: -8.9 QED: 0.47 SA: 0.44 / Vina: -8.9 QED: 0.43 SA: 0.72 / Vina: -8.9 QED: 0.22 SA: 0.68 / Vina: -8.6 QED: 0.54 SA: 0.50 / Vina: -8.6 QED: 0.74 SA: 0.73 / Vina: -8.6 QED: 0.45 SA: 0.59 — Reference (4rlu) Vina: -8.4 QED: 0.58 SA: 0.88

Ours (3b6h) — Vina: -7.8 QED: 0.92 SA: 0.68 / Vina: -7.6 QED: 0.76 SA: 0.65 / Vina: -7.5 QED: 0.55 SA: 0.67 / Vina: -7.5 QED: 0.72 SA: 0.72 / Vina: -7.3 QED: 0.77 SA: 0.77 / Vina: -7.2 QED: 0.70 SA: 0.79 — Reference (3b6h) Vina: -6.4 QED: 0.50 SA: 0.77

liGAN (3b6h) — Vina: -7.2 QED: 0.62 SA: 0.65 / Vina: -6.6 QED: 0.31 SA: 0.73 / Vina: -6.6 QED: 0.56 SA: 0.76 / Vina: -6.5 QED: 0.43 SA: 0.66 / Vina: -6.4 QED: 0.34 SA: 0.57 / Vina: -6.2 QED: 0.56 SA: 0.64 — Reference (3b6h) Vina: -6.4 QED: 0.50 SA: 0.77

## A.2 Linker Prediction

We present more examples of linker prediction. Since the baseline model DeLinker is graph-based and does not generate 3D linker structures, to make the results of both methods visually comparable, we only show 2D molecular graphs. We randomly selected 5 cases from the testing set. For each case, we select 5 representative molecules, including the molecule with best similarity, the molecule with worst similarity, and 3 molecules between them. These examples evidence that our method is generally more likely to produce linkers that recover or resemble the original structure.

| | Fragments | Predicted | | | | | Reference |
|---|---|---|---|---|---|---|---|
| 4qlk | | Ours | | | | | |
| | | Sim: 1.00 | Sim: 0.97 | Sim: 0.83 | Sim: 0.64 | Sim: 0.63 | |
| | | DeLinker | | | | | |
| | | Sim: 0.99 | Sim: 0.66 | Sim: 0.58 | Sim: 0.50 | Sim: 0.42 | |
| 3tym | | Ours | | | | | |
| | | Sim: 0.55 | Sim: 0.54 | Sim: 0.48 | Sim: 0.41 | Sim: 0.38 | |
| | | DeLinker | | | | | |
| | | Sim: 0.58 | Sim: 0.47 | Sim: 0.44 | Sim: 0.39 | Sim: 0.36 | |
| 3nfb | | Ours | | | | | |
| | | Sim: 1.00 | Sim: 0.91 | Sim: 0.87 | Sim: 0.85 | Sim: 0.79 | |
| | | DeLinker | | | | | |
| | | Sim: 0.90 | Sim: 0.57 | Sim: 0.56 | Sim: 0.52 | Sim: 0.45 | |
| 4xli | | Ours | | | | | |
| | | Sim: 1.00 | Sim: 0.57 | Sim: 0.57 | Sim: 0.56 | Sim: 0.56 | |
| | | DeLinker | | | | | |
| | | Sim: 0.71 | Sim: 0.57 | Sim: 0.57 | Sim: 0.56 | Sim: 0.56 | |
| 4m7t | | Ours | | | | | |
| | | Sim: 0.84 | Sim: 0.79 | Sim: 0.67 | Sim: 0.64 | Sim: 0.63 | |
| | | DeLinker | | | | | |
| | | Sim: 0.67 | Sim: 0.58 | Sim: 0.56 | Sim: 0.53 | Sim: 0.50 | |

# B  Additional Model Details

## B.1  Sampling Algorithm

At the first step of molecule generation, there is no placed atoms in the binding site. To sample the first atom, we use Metropolis-Hasting algorithm to draw samples from the marginal distribution $p(\boldsymbol{r}|\mathcal{C}) = \sum_e p(e, \boldsymbol{r}|\mathcal{C})$ and select coordinate-element pairs that have highest joint probability. We draw 1,000 initial samples from the Gaussian mixture model defined on the coordinates of protein atoms, whose standard deviation is 1Å. The proposal distribution is a Gaussian with 0.1Å standard deviation, and the total number of steps is 500.

If there are previously placed atoms in the binding site, to accelerate sampling and make full use of model parallelism, we discretize the 3D space onto meshgrids. The resolution of the meshgrid is 0.1Å. We only discretize the space where the radial distance to some frontier atom ranges from 1.0Å to 2.0Å in order to save memory. Note that frontier atoms are predicted by the frontier network. Then, we evaluate the non-normalized joint probabilities on the meshgrid and use `softmax` to normalize them. Finally, we draw coordinate-element pairs from the normalized probability.

We use the beam search technique to generate 100 different molecules for each binding site, and we set the beam width to 300.

## B.2  Hyperparameters

The hyperparameters are shared across both molecule design and linker prediction tasks. For the context encoder, the neighborhood size of $k$-NN graphs is 48, the number of message passing layers $L$ is 6, and the dimension of hidden features $\boldsymbol{h}_i^{(\ell)}$ is 256. For the spatial classifier, the dimension of $\boldsymbol{v}$ is 128, and the number of aggregated nodes is 32. We train the model using the Adam optimizer at learning rate 0.0001. The batch size is 4 and the number of training iterations is 1.5 million, which takes about 2 days on GPU.