# OpenReview forum: "A 3D Generative Model for Structure-Based Drug Design"
_NeurIPS.cc/2021/Conference — NeurIPS 2021 Poster_

### Official Review · Reviewer_kPk1 · 2021-07-16

**Rating:** 6
**Confidence:** 4

**Summary:**

This paper addresses the task of 3D molecular generation within a 3D protein binding pocket. An autoregressive model is used to predict probability distributions of atom occupancies, which are iteratively sampled to construct a full molecular structure inside the binding site.

**Limitations And Societal Impact:**

Extensions are discussed; social impact is not.

**Main Review:**

The approach is presented clearly and is remarkably simple. The sampling approach provides a natural way to decode the continuous distribution into a discrete molecular structure. The sampling procedure for training of observing masked molecular structures and training the joint distribution through sampling both positive and negative spatial coordinates is procedurally simple (in a good way, in my opinion). The primary criticism of this work would be that it does not provide any significant machine learning advance and lacks quantitative comparisons to plausible alternative approaches, instead focusing exclusively on a comparison to ref [21]. However, I do think this is an interesting task from a domain perspective.

Contextualization

1.	What are the aspects of the approaches in refs. 8, 30, and 31 that render them unable to handle larger molecules? This does not seem to be a fundamental limitation of the method but perhaps a limitation of the evaluations in those original papers.

2.	The statement that 1D/2D methods are unable to be applied to structure-based design tasks seems to be an oversimplification. String-based and graph-based methods have been applied to the optimization of docking scores as a structure-based oracle function (e.g., https://pubs.acs.org/doi/10.1021/acs.jcim.0c00833)  While not conditioning generation on a 3D structure explicitly, this may be considered an application to SBDD.

3.	There is a family of methods designed for atom-by-atom or fragment-by-fragment ligand generation within a fixed protein pocket (e.g., AutoGrow4). Is it feasible to do any performance comparisons to these non-ML methods? The problem formulation is different, as they use Vina Scores as intermediate signals during generation, but it is realistic to have access to a docking oracle function. The procedure is similar enough to this work that a comparison is warranted.


Experiments

4.	The performance of the model (i.e., Vina Scores) is not reported when the force field refinement is omitted. This paper does not state whether the force field refinement operates on the ligands alone or the ligands and the binding site together, but the reference implies the latter. Ref 21 also states that Vina was used for reoptimization (including of the pose), whereas this work implies that only the Vina scoring function was used. In the evaluation workflow, were structures taken as their 3D conformations or re-optimized? Insufficient details are provided with respect to this evaluation and the dataset used, although the authors do state that code/data will be made available upon publication.

5.	In the experiments, evaluation of activity focuses on the average and median values. However, the goal would arguably be to minimize the docking scores. It appears as though the model does not have the ability to learn how to generate the “most favorable” ligand by considering docking scores during training, but merely generate a representative ligand. Most generated molecules have binding affinity scores that are worse than the reference ligand.

6.	It seems surprising that in the top row of Figure 4, so many of the generated compounds have the same heterocyclic motif as the reference compound, not to mention the terminal phosphate group. Did this scaffold appear in the training set for a very similar protein? Or did the model truly rediscover it from scratch?


**Time Spent Reviewing:**

3

---

> ### Author Response · Authors · 2021-08-10
> **Author Response**
>
> We sincerely thank the reviewer for the careful and valuable comments. Before addressing the reviewer’s concern, we would like to discuss the spirit of our paper.
>
> We have noticed that the reviewer has focused much on specific metrics (such as the Vina score) --- the reviewer raised concerns about metrics (especially Vina score) in contextualization, experiments, and model design. These comments are very valuable and constructive, and we would like to thank the reviewer again for the comments.
>
> However, it is also worth noting that these metrics are *not the golden standard* for assessing drug molecules. If there are computable metrics that can serve as the golden standard for drugs, drug development won't be as difficult as it is now.
> In this regard, our model *doesn’t mean to* optimize for these man-made metrics (e.g. Vina, QED, SA, etc.), in contrast to most of the previous models, either ML-based or non-ML-based.
> Instead, our model focuses on how drugs work physically via explicitly learning to model the shape and interactions in the 3D space, where real-world proteins and molecules actually exist.
>
> Nevertheless, we are not repudiating the value of the widely used metrics --- they are still some kind of silver standard. In our context, the purpose of these metrics is to *reflect* the quality of the model's generated molecules. After all, these metrics are familiar to the community, and they can at least to some extent reflect the generation quality when there is no better choice.
>
> We understand it is possible that we haven’t conveyed the spirit of our paper clear enough. We will further clarify it in the revised version and we would like to thank the reviewer again for the precious comments.
>
> ---------
>
> **[Q] About force field refinement and Vina score calculation.**
>
> The purpose of force field refinement is to fix less accurate sub-structures. For example, benzene rings might be not perfectly planar, and some bond lengths might deviate a bit from the optimal value. If we feed the unrefined molecules to Vina, the subtle deviation in local structures would dominate Vina score, and in this case, Vina score less reflects how well the molecule binds to the pocket but pays more attention to the subtle structural deviation, which is unintended. Therefore, we believe it is safe to omit the Vina score without force field refinement.
> We follow the same protocol with Ref.21 when calculating Vina scores, which includes reoptimization. We are aware that the docking pose might be different from the generated pose, so we calculate the RMSD (without rotation and translation alignment) between the generated pose and the docked pose of high affinity ligands. 42.55% of the generated ligands have an RMSD less than 2Å, indicating reasonable poses of the generated molecules. We will include more details about evaluation in the revised version.
>
> ---------
>
> **[Q] About considering docking scores during training.**
>
> As discussed above, our model aims at learning to model the shape and interactions explicitly in the 3D space, rather than optimizing for some man-made metrics, different from almost all the previous molecule generative models. We report the Vina score because it can to some extent reflect the quality of generation, but we do not optimize our method for it.
> However, we agree that it would be valuable to design a model that considers both 3D shapes and scoring, which deserves future investigation.
>
> ---------
>
> **[Q] About dataset split and novelty of generated molecules.**
>
> We randomly drew the training set and test set from two clusters constructed by mmseqs. The sequence identity between these two clusters is less than 30%, so the proteins in the test set are not similar to those in the training set.
> To demonstrate that the model can build novel molecules rather than just remembering the molecules in the training set, we report the novelty score based on nearest neighbor similarity [16] of the generated molecules, which measures the percentage of generated molecules that are not similar to any molecule in the training set:
>
> |                 | all generated | high affinity |
> | :-------------: | :-----------: | :-----------: |
> | **Novelty (%)** |    73.94%     |    74.28%     |
>
> As for the top row of Figure 4, the molecule is an ADP-like compound, which is prevalent in the real world and in the database. Therefore, it is not surprising that the model identifies the binding sites for ADP and recovers an ADP-like molecule.
>
> ---------
>
> **[Q] About related methods.**
>
> We will revise the discussion on related methods according to the reviewer’s suggestion.
> The key difference between our method and previous 1D/2D methods is that our method directly models the shapes and interactions in the 3D space, while previous methods optimize molecules for an oracle docking score. AutoGrow4 belongs to this class of method, because it represents molecules as 2D graphs and optimizes molecules for Vina score, unable to explicitly perceive the 3D structure of binding sites. AutoGrow runs iteratively to optimize molecules, while our model generates ligands within a single sampling procedure. OptiMol can only be trained for specific pockets and cannot generalize across different pockets, while our model can generalize.
> Although it is generally unfair to compare our methods and previous ones using the metrics that previous methods optimize for, we will consider including comparison in the subsequent version.
>
> We would like to thank the reviewer again for the valuable comments, and  hope the above response could address the reviewer’s concern.
>
> ---------
>
> **[References]**
>
> - [16] Wengong Jin, Regina Barzilay, and Tommi Jaakkola. Composing molecules with multiple property constraints. arXiv preprint arXiv:2002.03244, 2020.

---

> > ### Comment · Reviewer_kPk1 · 2021-08-14
> > **Reply**
> >
> > I appreciate the thorough response!
> >
> > I would like to push back on the authors a bit regarding the value of the Vina metric. Here, they discuss how it is an artificial metric that vastly oversimplifies "drug discovery" (true). However, in response to a different reviewer, they explicitly state that "One of our goals is to generate molecules that bind to the pocket better than the reference molecule in the dataset. Therefore, we use the Vina score of the reference molecule as the criterion." As the authors have created this task more-or-less from scratch, it is important to be clear about how quantitative performance is meant to be evaluated and to either commit to the metrics they have selected, or propose better ones.
> >
> > I still have some concern with regards to the specific evaluations chosen and the lack of comparisons to methods that are relevant for the same task. Regarding comparisons, it is stated that "The key difference between our method and previous 1D/2D methods is that our method directly models the shapes and interactions in the 3D space, while previous methods optimize molecules for an oracle docking score." This seems to be comparing a _method_ to an _application/task_.
> >
> > I do think this is an interesting setting and trust the authors will revise the manuscript to provide the elaborations requested by the reviewers. I will raise my score above the accept threshold

---

> > > ### Author Response · Authors · 2021-09-20
> > > **We appreciate the reviewer’s valuable comments**
> > >
> > > We sincerely appreciate the reviewer’s constructive comments!
> > >
> > > We agree that our present work has some limitations regarding metrics and other aspects. We will discuss more about the limitations in the next version and identify directions for future work.
> > >
> > > As for the comparison, the ultimate goal of our method and other mentioned methods (OptiMol, MolGrow, etc.) is designing molecules that bind to proteins. Our method approaches this goal in a geometrical way. MolGrow, OptiMol and others approach the goal via a surrogate objective (e.g. Vina). Both ours and others try to approach the same ultimate goal, but formulate the problem differently.
> > >
> > > Again, we heartily appreciate the reviewer’s comment and the time spent on helping us improve the work. Thank you!

---

### Official Review · Reviewer_jsjN · 2021-07-16

**Rating:** 6
**Confidence:** 2

**Summary:**

This paper tackles the problem of generating molecules that bind to a specific protein. To make use of the spatial information to target the proteins exactly in 3D space, the paper proposes an autoregressive model where the atoms are sampled sequentially from a learned distribution that estimates the probability of atoms' occurrence in 3D space. In fact,   the paper models the probability of a position in 3D space being occupied by a certain type of atom given as input the 3D coordinate and the binding site.  the model encodes the context which is the binding site and concatenates it with the input query position r  and finally predicts the type atom that could occur at the position r.

**Limitations And Societal Impact:**

Limitation: the experiment did not include enough baselines that tackle similar problems.

**Main Review:**

The paper is well motivated and clearly written. They tackle the problem of structure-based drug design which is a very interesting and crucial step for using ml models for drug design.  The experiment results show clear improvement over the baselines.

There are few concerns
1. They only include one single baseline but I am sure there are many works that exist in this line and should be taken into consideration. For instance the work [30] and [31].

2.  In table 1, it is shown that the proposed model has a better score of not only Vina but also SA score and QED score. I can understand the model could lead to a better Vina score as the model is tailored to solve this specific problem but it is not clear what factor of the model leads to a better SA score and QED score when compared to the baseline.

**Time Spent Reviewing:**

4 hours

---

> ### Author Response · Authors · 2021-08-10
> **Author Response**
>
> We sincerely thank the reviewer for the constructive comments. Below is our response to the reviewer’s concern.
>
> **[Q1] About the baselines.**
>
> The work [30] and [31] also generate molecules directly in 3D space. However, they are not readily applicable to generating molecules for protein binding sites. To make them suitable for this task, one should at least: (1) design a new reward function that measures how well the molecule binds to the pocket, (2) design a very different network architecture that can perceive the 3D protein environment, and (3) design a different policy network that places atom not only according to previously generated atoms but also the protein environment. These modifications would lead to a significantly different new work. Therefore, [30] and [31] are not directly comparable with our model. Other related methods such as [8] are not directly applicable to the task either. However, the above-mentioned modifications to previous 3D-based molecule generation models would be valuable future work. We will add more discussion about these related methods in our revised version and we thank the reviewer for raising the concern.
>
> **[Q2] About factors leading to better SA and QED scores.**
>
> Our model generates less unrealistic molecules and less erroneous substructures, which greatly improves SA (more synthethable) and QED (more like drugs) scores. This can be evidenced in the visual comparison in Sec.A.1 of the supplementary material.
> We believe this is because our model represents molecules as continuous probability densities in the 3D space, and the densities are equivariant to the rotation and translation of the protein. In contrast, the baseline method uses 3D CNN to represent molecules. Due to the limited resolution of CNN and the lack of rotational equivariance, the quality of generated molecules is limited. We will add more discussion about SA and QED scores in the revised version.
>
> We would like to thank the reviewer again for the comments, and  hope the above response could address the reviewer’s concern.
>
> **[References]**
>
> - [8] Niklas WA Gebauer, Michael Gastegger, and Kristof T Schütt. Symmetry-adapted generation of 3d point sets for the targeted discovery of molecules. arXiv preprint arXiv:1906.00957, 2019.
> - [30] Gregor Simm, Robert Pinsler, and José Miguel Hernández-Lobato. Reinforcement learning for molecular design guided by quantum mechanics. In: International Conference on Machine Learning, pages 8959–8969. PMLR, 2020.
> - [31] Gregor NC Simm, Robert Pinsler, Gábor Csányi, and José Miguel Hernández-Lobato. Symmetry-aware actor-critic for 3d molecular design. arXiv preprint arXiv:2011.12747, 2020.

---

> > ### Comment · Reviewer_jsjN · 2021-08-25
> > **Thank you for the response**
> >
> > I would like to thank you for the author's response.  After reading other reviewers' comments and authors' response, I have a better understanding of the paper.
> >
> >  I decide to raise my score to 6 but  I do agree with Reviewer kPk1 comment on the "it is important to be clear about how quantitative performance is meant to be evaluated and to either commit to the metrics they have selected, or propose better ones". It is true (and authors also emphasized that ) the advantage of the model is that it models directly model the shape and interactions in the 3D space, it does not mean to optimize towards VIna score. But is not vina score is a reflection of that interaction between protein and molecule? and since the model has more advantage over those 2D/1D methods that only optimize towards such oracle score, should not the paper with more baselines of such kind and show experimental improvements? I understand the comment of the auther on the pointed two baselines, but I believe there are other works that could be used as baselines.

---

> > > ### Author Response · Authors · 2021-09-19
> > > **We appreciate the reviewer’s constructive response**
> > >
> > > We appreciate the reviewer’s constructive response!
> > >
> > > As pointed out by the reviewer, Vina score is a reflection of the interaction between protein and ligand, so we use Vina score as a metric for evaluation, but we do not design a model that optimizes for Vina scores.
> > >
> > > There are other methods for structure-based molecule generation such as MolGrow, OptiMol (pointed out by reviewer kPk1). However, as discussed in the response to reviewer kPk1, these methods have some other inherent limitations in addition to not being able to capture 3D structures, such as they cannot generalize across different proteins. We believe it is not easy to make direct comparisons, but we agree that it is interesting to compare them using Vina scores anyway.
> > >
> > > We find there are only a limited number of previous methods that we can compare our method to. Therefore, we believe we are trying to solve a valuable problem that hasn’t drawn enough attention previously. We hope our work can attract more researchers to this problem and set up a baseline for future work, further pushing the boundary of computer-aided drug design.
> > >
> > > Finally, we thank the reviewer again for the valuable comments!

---

### Official Review · Reviewer_RaVz · 2021-07-17

**Rating:** 7
**Confidence:** 4

**Summary:**

This paper introduces a novel generative method for molecular design that uses 3D spatial information and the binding pocket embedding to generate molecules with high binding affinity towards a selected target protein. The method allows for multimodal sampling of valid and diverse molecules. In the experiments, the model is evaluated in two scenarios. The first scenario is a structure-based de novo drug design, and the second one is linker prediction. In both cases, the proposed model manages to generate reasonable compounds that fit into the binding pocket. Additionally, the generated compounds exhibit high drug-likeness and synthetic accessibility.

**Limitations And Societal Impact:**

The authors could elaborate more on the limitations of the method, e.g. what the success rate of sampling molecules is or what is the time complexity of the method considering the sampling methods used. The negative societal impact is not described. The presented method is a general method for designing active compounds, so potential societal impacts may include improving drug discovery pipelines but also acceleration of the design of harmful compounds, e.g. toxic psychoactive substances.

**Main Review:**

The method presented in the paper is very original. The model can perceive 3D relation between generated ligand and target protein. There are not many generative models that are designed to create molecules that directly fit the 3D space of the binding pocket. The proposed model is certainly unique in the category of structure-based generative models Another method in this category is the shape-based model by Skalic et al. [1], which is missing in the references. This method generates compounds based on the predicted 3D shape that fits the binding pocket. However, also this method uses the SMILES representation instead of working directly on 3D atoms.

The presented experimental results are convincing. The authors present qualitative results in Table 1 which could be improved by adding more models to the comparison (e.g. [1]) and adding error bars based on multiple runs of the models. The qualitative results in Figures 4 and 5 clearly show that the proposed method can generate reasonable compounds. What seems a little bit suspicious is that in Figure 4 the generated compounds look very similar to the reference compound. It may suggest data leakage, e.g. compounds that are very similar to the reference one (e.g. the same analog series) are included in the training set. A scaffold split would prevent this from happening. To make these experiments more convincing, the whole protein families that are present in the testing set should be also discarded from the training set.

The paper is clearly written, and there are only minor issues that generate confusion. In Equation 1, the notation $W^\ell(d_{ij})$ may suggest that $W^\ell$ is a function, and in the text it is said that “weight of message from $j$ to $i$ depends only on $d_{ij}$”. On the other hand, this function is not defined, but instead it is said that $W$ is a learnable matrix. Equations 11 and 12 in the supplemental material are much more precise and do not seem equivalent to what is presented in the paper. The architecture of the frontier network $F$ is not described at all.

The proposed method supports multi-modal sampling of valid compounds thanks to a simple (not using a latent space) autoregressive method. The experiments show that the proposed generative model creates compounds that are drug-like and synthetically accessible without directly optimizing for these values. Additionally, a linker prediction task is performed and the results are compared with a popular model for this task, DeLinker. Having considered all that, the method presented in the paper is of high value to the field of computer-aided drug discovery.

Other comments:
- Why is ‘Nothing’ not a separate class in the probability vector $c$, but instead it always artificially gets value 0 by the means of Equation 5? Does the loss in Equation 8 not converge otherwise?
- In Figure 5, the PDB ID is missing.
- In Figure 5, the atoms with a red overlay are hardly readable.
- In Figure 3, it is not clear what the vertical dashed lines signify.

Based on all the above, I am leaning toward the acceptance of this paper.

[1] Skalic, Miha, et al. "Shape-based generative modeling for de novo drug design." Journal of chemical information and modeling 59.3 (2019): 1205-1214.

**Time Spent Reviewing:**

4

---

> ### Author Response · Authors · 2021-08-10
> **Author Response**
>
> We sincerely thank the reviewer for the careful and constructive comments.
>
> **[Q1] About dataset splits.**
>
> We randomly drew the training set and test set from two clusters constructed by mmseqs. The sequence identity between these two clusters is less than 30%, so we believe the proteins in the test set are not similar to those in the training set at the sequence level. We feel sorry about the confusion caused by the ambiguous description. We will clarify the protocol of making dataset splits in the revised version.
> We would also consider splitting the dataset according to ligand similarity. For now, we would like to report the novelty score based on nearest neighbor similarity [16] of the generated molecules, in order to demonstrate that the model can build novel molecules rather than just remembering the molecules in the training set.
>
> |                 | all generated | high affinity |
> | :-------------: | :-----------: | :-----------: |
> | **Novelty (%)** |    73.94%     |    74.28%     |
>
> As for the top row of Figure 4, the molecule is an ADP-like compound, which is prevalent in the real world and in the database. Therefore, it is not surprising that the model identifies the binding sites for ADP and recovers an ADP-like molecule.
> Based on these reasons, we believe there isn’t data leakage in our current dataset split. We will clarify the split details in the revised version.
>
>
> **[Q2] About the description of the network architecture.**
>
> We are sorry about the confusion caused. Wℓ(dij) is a weight network which takes distance dij as inputs and outputs weights, which is then used to multiply hidden embedding h to update it. It is similar to the definition of SchNet. We will double check the manuscript,  and add more necessary network architecture illustrations in the main text or supplementary materials. The models and datasets will also be made open-source once this paper is made public.
>
>
> **[Q3] Other issues.**
>
> - Why is “Nothing” a separate class: It is one way of parameterization. We believe that representing “Nothing” using a separate class also works well.
> - Missing references: We thank the reviewer for referring to related work, we will add the missing references and discuss them.
> - Figure 5: The PDB IDs are 3nfb and 4qlk respectively. We will add the missing PDB IDs in the revised version. We will also increase the transparency of the red overlay to enhance readability.
> - Figure 3: The dashed lines indicate a good score, though the criterion of high affinity varies across pockets. We will clarify the definition of dashed lines in the revised version.
> - Limitations and impacts: We thank the reviewer for summarizing the limitations and impacts of our work, we will add more discussion accordingly.
>
> We would like to thank the reviewer again for the comments, and hope the above response could address the reviewer’s concern.
>
>
> **[References]**
> - [16] Wengong Jin, Regina Barzilay, and Tommi Jaakkola. Composing molecules with multiple property constraints. arXiv preprint arXiv:2002.03244, 2020.

---

> > ### Comment · Reviewer_RaVz · 2021-08-17
> > **Thank you for the response**
> >
> > I would like to thank the authors for the detailed response.
> >
> > I have one more thought after reading the answers and other reviews. Reviewer kPk1 also noticed that the compounds in Figure 4 are very similar to the reference compound. It would be interesting to see the Tanimoto similarity between generated compounds and the most similar compound in the training dataset. As the authors point out, the dataset may contain many ADP-like (or GDP-like?) compounds. At the same time, other generative methods that generate molecules atom by atom oftentimes struggle to construct correct heterocycles, let alone a guanine which was reconstructed in 5 out of 6 generated compounds. I am inclined to suspect that the method could memorize more than just this structure from the dataset. The similarity score would confirm the novelty of these compounds. Also, Reviewer jsjN noticed that SA scores are high even though the model was not optimized for this property. This again could be caused by memorization of the structures in the dataset since the problem of synthetic accessibility is very complex. If I had to guess, I would say that SA scores should be positively correlated with the similarity to the closest compound in the training set, but better docking scores can be achieved only for more novel structures. However, this is only a loose thought/suggestion and not a criticism of the method.
> >
> > Thank you again for the response. I look forward to read the revised version if the paper is accepted.

---

> > > ### Author Response · Authors · 2021-09-18
> > > **We appreciate the reviewer’s insightful response**
> > >
> > > We deeply appreciate the reviewer’s insightful response! We will include more discussion and experiments that appeared in our previous response in the revised version.
> > > As pointed out by the reviewer, our model constructs heterocycles while other atom-based models often struggle to do so. We are also surprised by it early. But after observing the sampling process (see how the model places atoms step by step), we find that geometric information might play an important role in constructing cycles. For example, suppose we have placed 3 atoms A, B, and C, and suppose the angle $\langle \vec{BA}, \vec{BC} \rangle$ is approximately 108 degrees (the degree of regular pentagon’s inner angles). Since our model is 3D-based and can perceive geometric structures, it is likely that the model learns to place the remaining 2 atoms in order to complete a pentagon (5-cycle) given 3 already placed vertices. Other atom-based models operate using graph representation without 3D structures, so they use less information and there is no geometric constraints. We will consider discussing this point more in the revised version.
> > >
> > > The reviewer helps us a lot in improving the work. Again, we are grateful to the reviewer for the constructive comments!

---

### Official Review · Reviewer_GUvn · 2021-07-17

**Rating:** 7
**Confidence:** 3

**Summary:**

The paper proposed a deep generative model to design novel molecules in 3D space that bind to specific targets. Concretely, given the binding site as context, the generative model estimates the probability density of atoms’ occurrences in 3D space. Then after learning the distribution of atoms, they design an auto-regressive sampling scheme to generate atoms sequentially. Experimental results verifies the effectiveness of the proposed generative model in designing 3D molecules for specific proteins in binding pockets.

**Main Review:**

The paper designs a deep generative model to form a valid drug-like molecule fitting to a
specific binding site. The generation procedure is split into two parts: (i) it predicts
the probability density of atom occurrence in 3D space of the binding site, where the binding target is used as context and roto-translation invariance neural network is used; (ii) it leverages auto-regressive sampling algorithm for generating valid and multi-modal molecules from the learned atoms’ density. 3D molecule design is an important and popular topic in recent years and is highly valuable.

However, there are some issues in experiments.

First, the docking score (in terms of Vina) of the generated molecules is relatively poor. In Figure 4 and 5, none of the docking scores is better than -10. Also, in Table 1, the average vina score is -6.3, which does not indicate high affinity.

Second, during empirical studies, some important baseline methods are missing, e.g., [Masuda et al, Generating 3d molecular structures conditional on a receptor binding site with deep generative models] and [Ragoza et al. Learning a continuous representation of 3d molecular structures with deep generative models].


==================

I have read authors' responses and other reviews, and do not change my score.


**Time Spent Reviewing:**

3

---

> ### Author Response · Authors · 2021-08-10
> **Author Response**
>
> We sincerely thank the reviewer for the constructive comments. Below is our response to the reviewer’s concern.
>
> **[Q1] About Vina scores.**
>
> It is worth noting that the Vina scores of reference (“ground truth”) molecules in the test set are rarely better than -10 either (see the top-left histogram in Figure 3). The average Vina score of reference molecules is -7.2 according to Table 1. However, these reference molecules bind well to the pocket, even though their docking scores are worse than -10.
> One of our goals is to generate molecules that bind to the pocket *better* than the reference molecule in the dataset. Therefore, we use the Vina score of the reference molecule as the criterion --- If the generated molecules have better Vina score than the reference one, we consider it a “High Affinity” one. We reported the average percentage of molecules that have better Vina score than the reference molecule across all the pockets in the test set in terms of Vina scores. The result in Table 1 shows that our model is capable of generating molecules that bind to the pocket better than the reference molecule in the dataset.
> We thank the reviewer for pointing out this point that might lead to confusion. We will clarify it and discuss more about Vina scores in the revised version.
>
>
> **[Q2] About the baseline.**
>
> The baseline in [21] is called “liGAN” in the paper, which is the name of its open source repository on GitHub. We did compare our method with “liGAN” (see Tab.1, Fig.3, and supplementary material). We are sorry about using a confusing name for [21], and we will make it clear in the subsequent version.
> The model in [25] is not designed for generating molecules for protein binding sites. It is a general model that maps 3D molecules to a latent space and generates molecules by sampling in the latent space.
>
> We would like to thank the reviewer again for the comments, and hope the above response could address the reviewer’s concern.
>
>
> **[References]**
>
> - [21] Tomohide Masuda, Matthew Ragoza, and David Ryan Koes. Generating 3d molecular structures conditional on a receptor binding site with deep generative models. arXiv preprint arXiv:2010.14442, 2020.
> - [25] Matthew Ragoza, Tomohide Masuda, and David Ryan Koes. Learning a continuous representation of 3d molecular structures with deep generative models. arXiv preprint arXiv:2010.08687, 2020.

---

### Decision · Program_Chairs · 2021-09-27

**Decision:**

Accept (Poster)

**Comment:**

The AC and reviewers all agree that this is an interesting submission.

We strongly urge the authors to incorporate their clarifying comments into the manuscript. In addition, as mentioned by several reviewers, it is important that the authors be clear about the relative value of Vina score. As noted by kPk1, coming up with a new metric that is more appropriate would be a valuable direction for future study.

Reviewer RaVz made some very pertinent point on Tanimoto similarity, SA scores...etc. It would be very exciting to see these comments addressed either in the present manuscript or in future work.